# Concatenated modular BK channel constructs reveal divergent stoichiometry in gating control by LRRC26 (γ1), pore, and selectivity filter

Guanxing Chen, Qin Li, Kunal Shah, Jiusheng Yan*

Department of Anesthesiology and Perioperative Medicine, The University of Texas MD Anderson Cancer Center, Houston, United States

*For correspondence:
jyan1@mdanderson.org

## eLife Assessment

In this **important** contribution, Yan and colleagues describe a powerful and **compelling** strategy to generate concatamers of the BK channel and their fusion constructs with the auxiliary gamma subunits, which allows exploring contributions of individual subunits of the tetrameric channel to its gating and the study of heteromeric channel complexes of defined composition. Distinct examples are presented, which illustrate great diversity in the stoichiometric control of BK channel gating, depending on the site and nature of molecular perturbations. The molecular approaches could be extended to other membrane proteins whose N and C termini face opposite sides of the membrane.

**Abstract** Big-conductance, $Ca^{2+}$-activated $K^+$ (BK) channels consist of $Ca^{2+}$- and voltage-sensing, pore-forming α (BKα) subunits and regulatory auxiliary β or γ subunits. Concatenated subunit constructs are powerful tools for elucidating subunit stoichiometry in ion channel gating and regulation, allowing control over subunit arrangement, stoichiometry, and mutation. However, the additional S0 transmembrane segment in BKα places its N- and C-termini on opposite sides of the membrane, preventing tandem BK channel subunit construction by conventional methods. To investigate the atypical 'all-or-none' modulatory function of γ subunits and the subunit stoichiometry of BK channel gating, we developed concatenated constructs containing 2 or 4 BKα subunits by splicing them into modular forms that can be co-expressed to form functional channels. These constructs retained voltage and $Ca^{2+}$ gating properties similar to intact BK channels. By fusing the LRRC26 (γ1) subunit to the N-terminus of tandem BKα constructs, we found that a single γ1 subunit per α subunit tetramer is sufficient to fully modulate the channel. Furthermore, the L312A mutation in the deep pore region exhibited a stoichiometrically graded effect on voltage-gated BK channel activation. In contrast, a V288A mutation at the selectivity filter induced channel inactivation only when present in all four BKα subunits. Thus, by engineering concatenated BKα constructs, we identified three distinct stoichiometric modes of BK channel gating control by LRRC26, the pore, and the selectivity filter. This study offers new molecular tools and advances our understanding of subunit stoichiometry in BK channel gating and modulation.

## Introduction

The big-conductance, calcium- and voltage-activated $K^+$ (BK) channel is a unique member of the potassium channel family, characterized by exceptionally large single-channel conductance and dual regulation by membrane voltage and intracellular free $Ca^{2+}$ (*Salkoff et al., 2006*). The BK channel is a

homotetramer composed of four identical pore-forming, $Ca^{2+}$- and voltage-sensing α (BKα) subunits (~130 kDa) and variable auxiliary subunits. BK channels exhibit prominent features in molecular architecture (*Tao and MacKinnon, 2019*; *Tao et al., 2023*) and allosteric gating mechanisms (*Horrigan and Aldrich, 2002*; *Xia et al., 2002*). In the transmembrane (TM) domains, BK channels differ from most voltage-gated $K^+$ (Kv) channels by possessing an extra S0 TM helix, lacking domain swapping between the S1-S4 voltage-sensing domain (VSD) and the S5-S6 pore-gate domain (PGD), and exhibiting tight VSD-PGD packing via extensive S4-S5 interactions (*Tao and MacKinnon, 2019*). The BKα subunit also contains a large cytosolic C-terminus composed of two tandem RCK domains (RCK1, RCK2) responsible for $Ca^{2+}$ and $Mg^{2+}$ sensing (*Xia et al., 2002*; *Yang et al., 2015*; *Wu et al., 2010*; *Hite et al., 2017*; *Tao et al., 2017*; *Shi et al., 2002*; *Yusifov et al., 2008*). The RCK domains from all BKα subunits assemble into a tetrameric two-layer gating ring that expands and shifts toward the membrane in response to $Ca^{2+}$ bindings, leading to S6 movement and channel opening (*Hite et al., 2017*; *Tao et al., 2017*).

BK channel function is regulated by auxiliary β and γ subunits and by LINGO1, conferring tissue-specific gating and pharmacological properties (*Solaro and Lingle, 1992*; *Wallner et al., 1995*; *Brenner et al., 2000*; *Yan and Aldrich, 2010*; *Yan and Aldrich, 2012*; *Guan et al., 2017*; *Dudem et al., 2020*). The four γ subunits (γ1-γ4), also known as LRRC26, LRRC52, LRRC55, and LRRC38, are leucine-rich repeat (LRR)-containing membrane proteins (*Yan and Aldrich, 2010*; *Yan and Aldrich, 2012*). The γ subunits facilitate BK channel activation by shifting the voltage dependence of channel activation in the hyperpolarizing direction by ~140 mV (γ1), 100 mV (γ2), 50 mV (γ3), and 20 mV (γ4), in terms of half-maximal activation voltage ($V_{1/2}$) in the absence of $Ca^{2+}$. The γ1 subunit likely modulates BK channels by enhancing the allosteric coupling between VSD activation and the pore opening (*Yan and Aldrich, 2010*). All γ subunits share a common topology, including an N-terminal signal peptide, an extracellular LRR domain, a single TM segment, and a short intracellular C-terminus (*Yan and Aldrich, 2010*; *Yan and Aldrich, 2012*; *Chen et al., 2022*). Their modulatory effects on BK channel voltage gating are mainly determined by the TM segments and C-terminal clusters of positively charged residues (*Li et al., 2016*; *Li et al., 2015*), while the LRR domains regulate the γ subunits' expression and surface trafficking (19). Recent cryo-EM structures of BKα/γ1 complexes show that the LRRC26's TM segment binds peripherally to the BKα VSD, involving S0, S2, S3, and pre-S1 helices, while the LRR domains tetramerize extracellularly without directly contacting BKα (*Kallure et al., 2023*; *Yamanouchi et al., 2023*; *Redhardt et al., 2024*).

A fundamental question in ion channel gating and regulation is the structural and functional subunit stoichiometry. BK channel modulation by the γ subunits exhibits an atypical binary 'all-or-none' phenotype: voltage-dependence ($V_{1/2}$) is either fully shifted or unchanged under limited γ1 expression (*Chen et al., 2022*; *Gonzalez-Perez et al., 2014*). In tetrameric ion channels, auxiliary proteins typically follow fourfold symmetry, producing graded modulation based on subunit stoichiometry relative to the pore-forming principal subunit, as observed with BK β subunits (*Wang et al., 2002*), KCNE subunits on KCNQ channels (*Nakajo et al., 2010*), and KChIP subunits on Kv4 channels (*Kitazawa et al., 2014*). Consistent with previous reports detecting up to four γ1 subunits per channel (*Gonzalez-Perez et al., 2018*; *Noda et al., 2020*; *Carrasquel-Ursulaez et al., 2018*), Cryo-EM structures also show a symmetric presence of four γ1 subunits in BKα/γ1 complexes (*Kallure et al., 2023*; *Yamanouchi et al., 2023*; *Redhardt et al., 2024*). However, single-channel recordings using a β2-γ1 chimeric subunit suggest that even a single γ1 subunit is sufficient for full modulation (*Gonzalez-Perez et al., 2018*). It is of note that the evidence remains inconclusive due to several limitations: the chimeric construct lacks the LRR domain, which may influence γ1 function (*Chen et al., 2022*); the inferred subunit number is indirectly based on the β2 N-terminal blockade effect; and the sample size in number of channels is limited by the single-channel recording method. Moreover, given the largely independent impact of the individual VSDs to BK channel gating (*Horrigan and Aldrich, 2002*) and the symmetric presence of γ1 near all VSDs (*Kallure et al., 2023*; *Yamanouchi et al., 2023*; *Redhardt et al., 2024*), an alternative concerted 'all-subunits-required' model, involving extracellular LRR domain tetramerization, has been proposed (*Kallure et al., 2023*). Therefore, direct biochemical determination of the functional stoichiometry of γ1 in BK channel modulation is needed.

Concatenated subunit constructs with 2 or 4 channel subunits fused together in a C-to-N-terminal arrangement have been powerful tools for dissecting subunit stoichiometry and cooperativity in various voltage- and/or ligand-gated channels (*Hurst et al., 1995*; *Fahlke et al., 1998*; *Minier and*

*Sigel, 2004*; *White, 2006*; *Ogielska et al., 1995*; *Zandany et al., 2008*; *Wu et al., 2014*). However, the additional S0 segment in BKα prevents straightforward concatenation, as its N- and C-termini reside on opposite sides of the membrane. BK channels, owing to their unique biophysical properties, serve as a value model for studying allosteric gating mechanisms of multimodal ion channels (*Horrigan and Aldrich, 2002*). Yet, the unavailability of functional BKα concatemers has hindered precise stoichiometric investigations of channel gating and modulation at both intra- and inter-subunit levels.

To address this, we engineered modular BKα constructs that reassemble into functional concatenated channels with biophysical properties comparable to intact BK channels. Using these, we demonstrate that a single γ1 subunit per BKα tetramer is sufficient to fully modulate the channel. Given the central role of the PGD in BK channel gating, we further applied this system to mutational analyses of the deep pore and selectivity filter. We revealed distinct stoichiometric requirements for gating control by LRRC26, the pore, and the selectivity filter. This study provides new molecular tools and mechanistic insights into the stoichiometry of BK channel gating and regulation.

## Results

### Construction of functional BK channels with concatenated tandem repeats of the α subunits

We employed multiple strategies to generate concatenated BKα subunit constructs that enable expression of functional channels with biophysical properties that are comparable to intact (i.e. unsplit) BK channels and facile amplification and manipulation at the plasmid DNA level. Given the extracellular location of the N-terminus of BKα (*Figure 1A*), which precludes direct C-to-N-terminal concatenation, we first attempted to split BKα into the N-terminal S0 part and the remaining major portion. This was based on the report that co-expression of these two parts can form functional channels (*Wallner et al., 1996*). However, plasmids of tandem constructs lacking only S0 proved difficult to generate and use due to large plasmid size and instability during cloning. Since the C-terminal RCK2 domain can also be expressed as a module that forms functional channels when split and co-expressed with the rest of BKα (*Wei et al., 1994*), we thus further split BKα into a main part (residues 94–649), designated as $\alpha_M$, and the rest by deletion of the main part, designated as $\alpha^{\Delta M}$ (*Figure 1A and B*). The $\alpha_M$ module contains the major TM region (S1 to S6) and the RCK1 domain. The $\alpha^{\Delta M}$ construct retains the N-terminal residues 1–93 including S0 and the C-terminal residues 652–1113 including the RCK2 domain.

To prevent homologous recombination within the concatenated tandem repeat constructs during molecular cloning and to facilitate site-directed mutations on specific subunits, we designed each BKα subunit in the repeats to have distinct DNA sequences. This was achieved by utilizing codon-optimized cDNA sequences for the 2nd, 3rd, and 4th repeats of the main part, which differ by ~25% in nucleotide sequence from the original 1st repeat and from each other. With these strategies, we generated a single unit and tandem constructs of double and quadruple repeats of $\alpha_M$, named $\alpha_{M(mono)}$ ($\alpha_M$), $\alpha_{M(dual)}$ ($\alpha_{MM}$), and $\alpha_{M(quad)}$ ($\alpha_{MMMM}$), respectively (*Figure 1B*). To evaluate the expression and stability of the concatenated tandem $\alpha_M$ constructs, we performed immunoblot analysis of the C-terminally V5-tagged constructs transfected in HEK293 cells. The result showed predominant protein bands of $\alpha_{MM}$ and $\alpha_{MMMM}$ at expected protein sizes recognized by the anti-V5 antibody (*Figure 1C*), confirming the expression and lack of major degradation of the concatenated $\alpha_M$ constructs. The expression level of the $\alpha^{\Delta M}$ construct was affected by the $\alpha_M$ construct, and co-immunoprecipitation confirmed the complex formation between the $\alpha_M$ and $\alpha^{\Delta M}$ constructs (*Figure 1—figure supplement 1*).

By co-expression of the C-terminally GFP-tagged $\alpha^{\Delta M}$ ($\alpha^{\Delta M}$-GFP) with single, double, or quadruple repeat constructs of $\alpha_M$, we observed formation of functional BK channels that resemble the intact BKα channel in their voltage and $Ca^{2+}$-dependence of channel activation (*Figure 1D and E*). The $V_{1/2}$ values of the BK channels formed by the single-unit $\alpha_M$ construct, when co-expressed with $\alpha^{\Delta M}$-GFP, were 192 and 16 mV at virtual 0 and 10 μM $Ca^{2+}$, respectively. The tandem double repeat constructs, $\alpha_{MM}$, when co-expressed with $\alpha^{\Delta M}$-GFP, produced functional BK channels with $V_{1/2}$ values of 186 mV at virtual 0 $Ca^{2+}$, and 16 mV at 10 μM $Ca^{2+}$ (*Figure 1E*; *Table 1*). Furthermore, the channels formed by tandem quadruple repeat construct $\alpha_{MMMM}$ and $\alpha^{\Delta M}$-GFP had $V_{1/2}$ values of 184 and 29 mV at virtual 0 and 10 μM $Ca^{2+}$, respectively (*Figure 1E*; *Table 1*). These $V_{1/2}$ values of engineered BK channels formed by single, double, and quadruple repeat constructs of $\alpha_M$ are close to those of the intact BKα

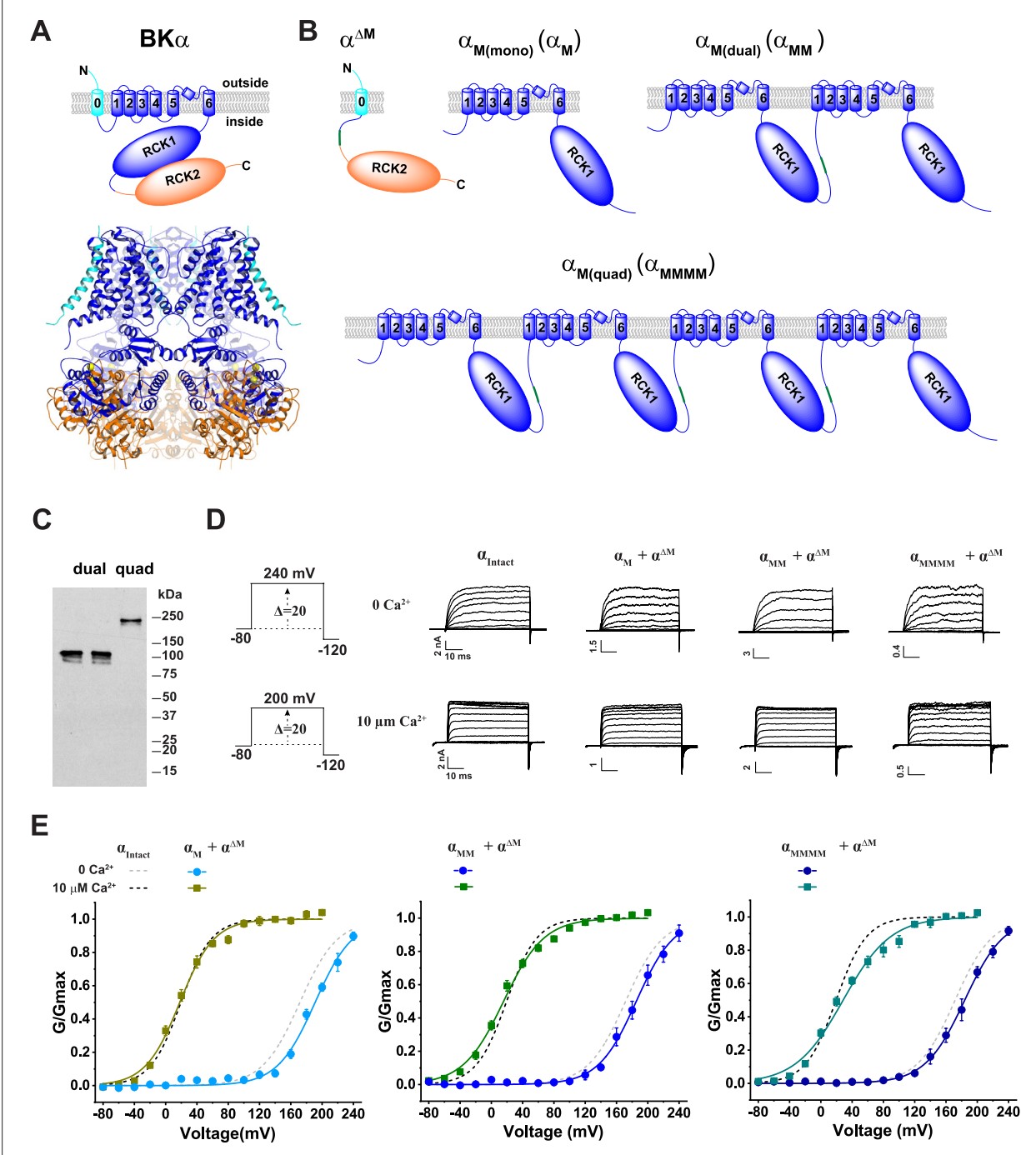

**Figure 1.** Design of concatenated BKα subunit constructs that form functional channels. (**A**) Schematic of the membrane topology and side view of the tetrameric 3D structure of the BKα subunit (PDB ID: 8GHF; cryo-EM structure in plasma membrane *Tao et al., 2023*) highlighting the three complementary separable regions in different colors. For clarity, the front and back subunits are shown in a partially transparent mode. (**B**) Schematic of the membrane topology for the BKα$_M$ module, concatenated dual- and quadruple-repeat constructs, and the complementary BKα$^{\Delta M}$ construct. (**C**) Immunoblot analysis of the V5-tagged α$_{M(dual)}$ (left, α$_{M1M2}$; right, α$_{M3M4}$) and BKα$_{M(quad)}$ constructs transiently expressed in HEK293 cells with an anti-V5 antibody. (**D**) Representative current traces from BK channels formed by intact, single repeat BKα$_M$, dual-repeat BKα$_{M(dual)}$, and quadruple-repeat BKα$_{M(quad)}$ constructs in response to membrane depolarization from −80 mV in 20 mV steps at 0 and 10 μM intracellular free Ca$^{2+}$. (**E**) Voltage dependence of BK channel activation for channels formed by the single (left), dual (middle), and quadruple (right) α$_M$ constructs in the absence and presence of 10 μM Ca$^{2+}$. Electrophysiological recordings were repeated n=4–10, as indicated in *Table 1*. Error bars represent ± SEM.

The online version of this article includes the following source data and figure supplement(s) for figure 1:

*Figure 1 continued on next page*

*Figure 1 continued*

**Source data 1.** Files containing original western blots for *Figure 1C*, indicating the relevant bands and treatments.

**Source data 2.** Original files for western blot analysis displayed in *Figure 1C*.

**Figure supplement 1.** Complex formation between split BKα subunit constructs.

**Figure supplement 1—source data 1.** Files containing original western blots for *Figure 1—figure supplement 1* left panel, indicating the relevant bands and treatments.

**Figure supplement 1—source data 2.** Original files for western blot analysis displayed in *Figure 1—figure supplement 1* left panel.

**Figure supplement 1—source data 3.** Files containing original western blots for *Figure 1—figure supplement 1* right panel, indicating the relevant bands and treatments.

**Figure supplement 1—source data 4.** Original files for western blot analysis displayed in *Figure 1—figure supplement 1* right panel.

channels, which had $V_{1/2}$ = 172 and 19 mV at virtual 0 and 10 μM $Ca^{2+}$, respectively (*Table 1*). These results show that the voltage- and $Ca^{2+}$-gating properties are largely unaltered in the engineered BK channels formed by engineered concatenated BKα subunit constructs.

## A single LRRC26 (γ1) subunit per channel is sufficient to fully modulate BK channels

The γ1 subunit's 'all-or-none' modulatory effect on BK channels has remained mechanistically elusive, despite the recent availability of 3D structures of the BKα/γ1 complexes. To directly investigate the functional stoichiometry of BK channel modulation by the γ1 subunit, we fused the C-terminus of the γ1 subunit to the N-terminus of the first $α_M$ repeat in the $α_{MM}$ and $α_{MMMM}$ constructs, generating the BKγ1$α_{MM}$ (BKγ1$α_{M(dual)}$) and BKγ1$α_{MMMM}$ (BKγ1$α_{M(quad)}$) fusion constructs (*Figure 2B*). Immunoblot analysis of the V5-tagged constructs using an anti-V5 antibody showed a major band at the expected size for both BKγ1$α_{MM}$ (~150 kDa) and BKγ1$α_{MMMM}$ (~270 kDa; *Figure 2C*), confirming proper expression and stability. Upon co-expression of these constructs with $α^{ΔM}$-GFP in HEK293 cells, we observed that the N-terminally fused γ1 subunit induced a $V_{1/2}$ shift of 125 mV with the BKγ1$α_{MM}$ construct ($V_{1/2}$ = 62 ± 5 mV) and 145 mV with the BKγ1$α_{MMMM}$ construct ($V_{1/2}$ = 38 ± 4 mV) in the virtual absence of $Ca^{2+}$ (*Figure 2D and E*). These large $V_{1/2}$-shifting effects in stoichiometrically defined γ1:$α$=1:2 and 1:4 channel complexes clearly indicate that a single γ1 subunit per tetrameric channel is sufficient to fully modulate BK channels. This 'one-subunit-sufficient' effect provides a mechanistic explanation for the observed binary, all-or-none modulation by the γ1 subunit when its expression is limited.

## L312A mutation reveals stoichiometrically graded gating and confirms functional integrity of concatenated constructs

To ensure that the concatenated $α_M$ constructs are suitable for studying the stoichiometry of BK channel gating and regulation, the constructs must demonstrate two additional aspects of functional integrity beyond simply forming functional channels. First, each $α_M$ repeat within the construct should contribute equally to channel formation. Second, the concatenated dual and quadruple $α_M$ repeats in the $α_{M(dual)}$ and $α_{M(quad)}$ constructs should assemble primarily as complete units, that is two $α_{M(dual)}$ constructs or one $α_{M(quad)}$ construct should form a single channel. The latter ensures that heterogeneous channels with uncontrolled subunit stoichiometry are unlikely to form and confound the results.

To evaluate the functional integrity of the concatenated constructs and to examine the stoichiometry of activation gating in the PGD, we investigated the mutational effects of the deep pore residue L312 ($^{307}$FILGGLAMFAS$^{317}$; *Figure 3A*), which plays a pivotal role in BK channel activation gating. Most mutations at this site resulted in constitutively active channels (*Chen et al., 2014*). The L312A mutation, in particular, causes a substantial shift of the $V_{1/2}$ toward hyperpolarization potentials in the absence of $Ca^{2+}$ (*Chen et al., 2014*; *Wu et al., 2009*). We first confirmed that the L312A mutation on the single-repeat $α_M$ construct caused a similarly large (~130 mV) shift in $V_{1/2}$ toward hyperpolarization in $Ca^{2+}$-free conditions (*Figure 3B and C*; *Table 1*). We then introduced the L312A mutation into the dual- and quadruple-repeat $α_{MM}$ and $α_{MMMM}$ constructs in a repeat-specific manner. For the $α_{MM}$ construct, introducing L312A mutation into either the first or second repeat resulted in a similar shift (65 or 68 mV) in $V_{1/2}$, approximately half of the total shift seen when both repeats (i.e., all subunits) were mutated (*Figure 3C*; *Table 1*). For the $α_{MMMM}$ construct, we observed that

**Table 1.** Boltzmann-fit parameters of the voltage-dependent concatenated tandem BK channel activation in the wildtype, mutants in the absence and presence of intracellular $Ca^{2+}$.

| Expression* | $Ca^{2+}$ (μM) | Boltzmann fit parameters | | |
| --- | --- | --- | --- | --- |
| | | $V_{1/2}$ (mV) | z | n† |
| $\alpha_{intact}$ | 0 | 172±2 | 1.08±0.04 | 9 |
| $\alpha_{intact}$ | 10 | 19±4 | 1.47±0.10 | 8 |
| $\alpha_M$ | 0 | 191±1 | 1.11±0.13 | 4 |
| $\alpha_M$ | 10 | 18±3 | 1.20±0.06 | 6 |
| $\alpha_{MM}$ | 0 | 186±5 | 1.12±0.12 | 5 |
| $\alpha_{MM}$ | 10 | 16±3 | 1.02±0.03 | 10 |
| $\alpha_{MMMM}$ | 0 | 184±5 | 1.09±0.07 | 6 |
| $\alpha_{MMMM}$ | 10 | 29±4 | 0.86±0.06 | 4 |
| $\alpha_{intact} + \gamma1$ | 0 | 35±2 | 1.56±0.11 | 8 |
| $\gamma1\alpha_{MM}$ | 0 | 56±4 | 0.99±0.07 | 6 |
| $\gamma1\alpha_{MMMM}$ | 0 | 42±3 | 1.13±0.13 | 5 |
| $\alpha_M^{L312A}$ | 0 | 62±4 | 0.96±0.07 | 6 |
| $\alpha_M^{L312A}{}_M^{WT}$ | 0 | 121±6 | 0.90±0.04 | 10 |
| $\alpha_M^{WT}{}_M^{L312A}$ | 0 | 118±5 | 0.95±0.07 | 4 |
| $\alpha_M^{L312A}{}_M^{L312A}$ | 0 | 64±5 | 0.97±0.08 | 5 |
| $\alpha_M^{L312A}{}_{M2}^{WT}{}_{M3}^{WT}{}_{M4}^{WT}$ | 0 | 153±4 | 0.96±0.10 | 5 |
| $\alpha_M^{L312A}{}_M^{L312A}{}_M^{WT}{}_M^{WT}$ | 0 | 133±2 | 0.77±0.05 | 4 |
| $\alpha_M^{L312A}{}_M^{L312A}{}_M^{L312A}{}_M^{WT}$ | 0 | 98±5 | 0.90±0.11 | 5 |
| $\alpha_M^{L312A}{}_M^{L312A}{}_M^{L312A}{}_M^{L312A}$ | 0 | 61±6 | 1.05±0.09 | 5 |
| $\alpha_M^{L312A}{}_M^{L312A}{}_M^{L312A}{}_M^{L312A} + \alpha_M^{WT}{}_M^{WT}{}_M^{WT}{}_M^{WT}$ | 10 | 180±13 (36%) / 65±8 (64%) | 0.90±0.52 / 1.01±0.13 | 5 |
| $\alpha_{intact}^{V288A}$ | 10 | 149±8 | 1.06±0.10 | 5 |
| $\alpha_M^{V288A}{}_M^{WT}$ | 10 | 31±2 | 1.14±0.09 | 5 |
| $\alpha_M^{V288A}{}_M^{WT}{}_M^{WT}{}_M^{WT}$ | 10 | 21±11 | 1.16±0.06 | 3 |
| $\alpha_M^{V288A}{}_M^{V288A}{}_M^{WT}{}_M^{WT}$ | 10 | 19±2 | 1.19±0.17 | 3 |
| $\alpha_M^{V288A}{}_M^{V288A}{}_M^{WT}{}_M^{V288A}$ | 10 | 38±3 | 1.05±0.06 | 3 |
| $\alpha_M^{V288A}{}_M^{V288A}{}_M^{V288A}{}_M^{V288A}$ | 10 | 127±10 | 0.75±0.21 | 3 |

*Except for the full-length BK$\alpha_{intact}$ construct, all other listed (derivatives of $\alpha_M$) constructs were co-expressed with the complementary $\alpha^{\Delta M}$-GFP construct.

†The number of recorded excised inside-out patches from different HEK293 cells.

the voltage dependence of the channel activation shifted progressively with each additional L312A mutation introduced. Specifically, the initial L312A mutation on the first repeat shifted $V_{1/2}$ by 31 mV, with further increases of 20, 35, and 37 mV in $V_{1/2}$ shifts observed for the additional mutation on the second, third, and fourth repeats, respectively (*Figure 3D*; *Table 1*). The L312A mutation also caused a slowed decay in tail currents at negative voltages (*Figure 3B*) compared to unmutated channels (*Figure 1D*). Fitting the current decay kinetics revealed that the time constant ($\tau$) of the decay was increased with the number of L312A mutations present in the channels formed by $\alpha_{MM}$ and $\alpha_{MMMM}$ constructs (*Figure 1E*). It is worth noting that the tail current decay rates for channels with zero or one L312A mutation were likely overestimated due to rapid closure at very negative voltages (−120 mV) in $Ca^{2+}$-free conditions, exceeding the detection limit of the 2 kHz-filtered recordings. These results

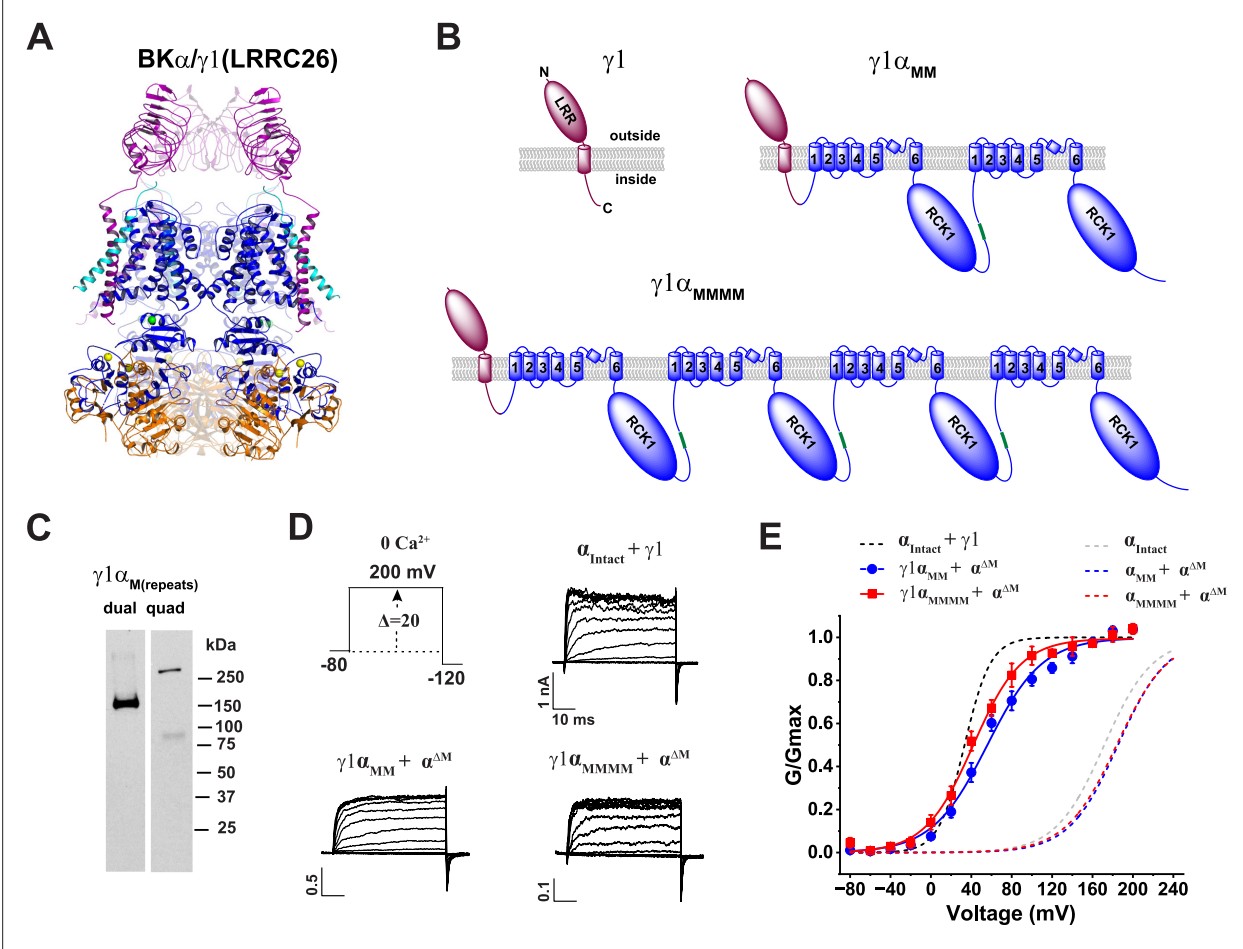

**Figure 2.** A single γ1 subunit per BK channel is sufficient for full modulation. (**A**) Side view of the 3D structure of the BKα/γ1 channel complex (PDB ID: 7YO3 *Yamanouchi et al., 2023*) showing the γ1 subunit in purple and the three separable BKα regions in distinct colors. (**B**) Schematic of the membrane topology for the γ1 subunit and its fusion constructs, created by linking its C-terminus to the N-terminus of the αM dual- and quadruple-repeat constructs. (**C**) Immunoblot analysis of the V5-tagged $γ1α_{M(dual)}$ and $γ1α_{M(quad)}$ constructs expressed in HEK293 cells with an anti-V5 antibody. (**D**) Representative current traces from BK channels formed by co-expressing the γ1 subunit with the intact BKα or by γ1-fusion to the concatenated $α_M$ constructs (co-expressed with $α^{ΔM}$) in response to membrane depolarization from −80 mV in 20 mV steps in the virtual absence of $Ca^{2+}$. (**E**) Voltage dependence of activation for channels formed by the $γ1α_{M(dual)}$ and $γ1α_{M(quad)}$ constructs co-expressed with $α^{ΔM}$. Electrophysiological recordings were repeated n=5–8, as indicated in *Table 1*. Error bars represent ± SEM.

The online version of this article includes the following source data for figure 2:

**Source data 1.** Files containing original western blots for *Figure 2C*, indicating the relevant bands and treatments.

**Source data 2.** Original files for western blot analysis displayed in *Figure 2C*.

demonstrate a stoichiometrically incremental effect of the L312A mutation on BK channel voltage gating. Importantly, they also indicate that individual $BKα_M$ repeats in both the dual- and quadruple-repeat constructs contribute similarly to the gating properties of the assembled channels.

The conductance-voltage (G-V) curves of BK channels formed by these L312A mutant constructs were well-fit by a single Boltzmann function, with slopes (i.e. apparent gating charge z) remaining within the typical range (*Figure 3C and D*; *Table 1*), consistent with a largely homogenous channel population. In contrast, G-V curves of the channels formed by co-transfecting cells with WT and L312A single BKα subunit constructs (1:1 DNA ratio) showed shallow slopes (*Figure 3F*), agreeing with the predicted heterogeneity in subunit composition and $V_{1/2}$ values. To confirm that each $α_{M(quad)}$ construct predominantly forms a single channel without subunit exchange, we co-expressed fully mutated $α_M^{L312A}{}_M^{L312A}{}_M^{L312A}{}_M^{L312A}$ and unmutated $α_M^{WT}{}_M^{WT}{}_M^{WT}{}_M^{WT}$ constructs (1:1 DNA ratio). The resulting G-V curves were best fit with a double-Boltzmann function showing $V_{1/2}$ values matching those of the all-WT and all-L312A channels (*Figure 3F*; *Table 1*). Additionally, tail currents at −120 mV exhibited

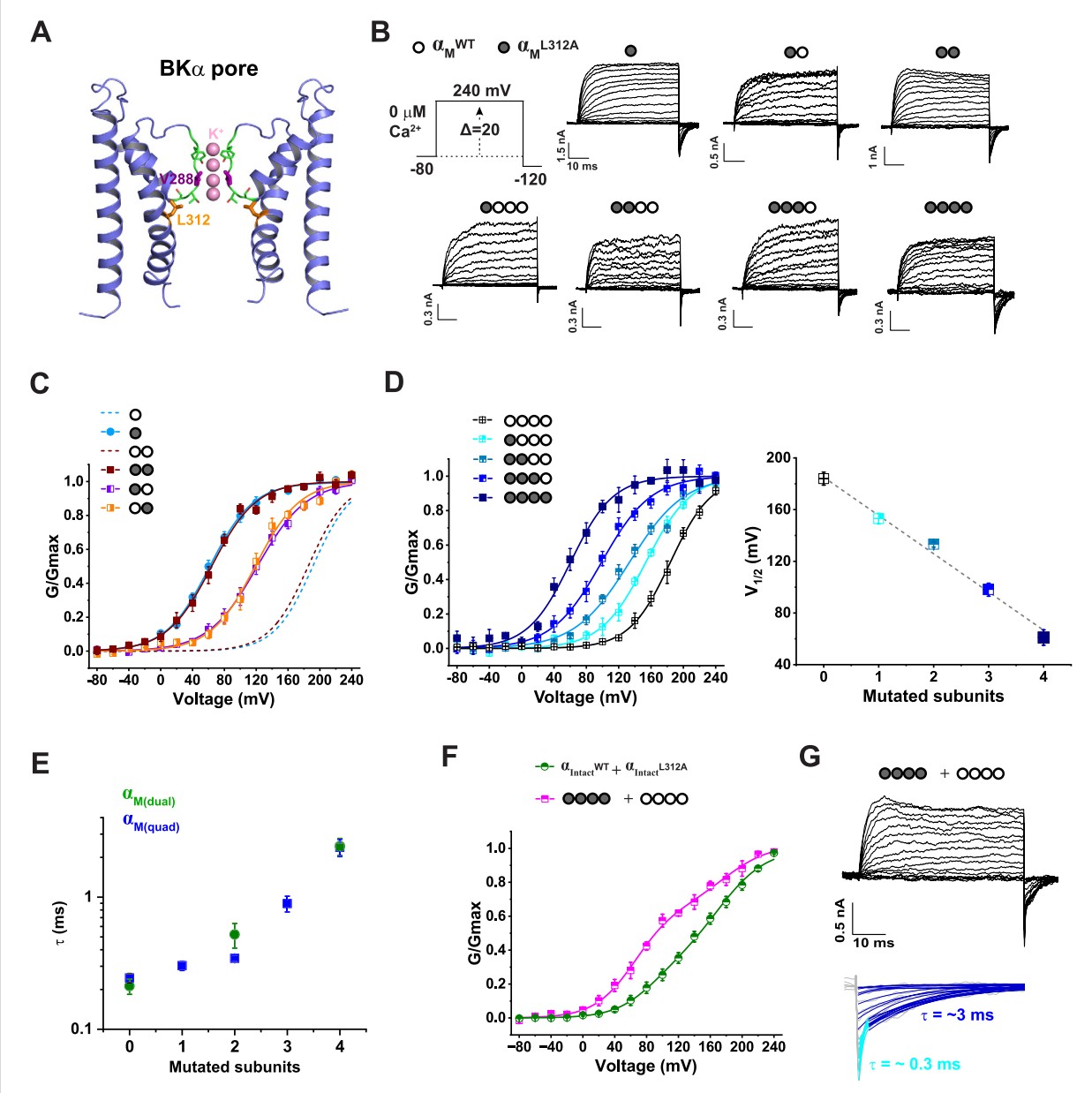

**Figure 3.** Stoichiometrically incremental effect of the L312A mutation on BK channel voltage gating and validation of functional integrity of the concatenated constructs. (**A**) Side view of the BK channel pore structure (PDB ID: 8GHF *Tao et al., 2023*), highlighting the deep pore residue L312 and selectivity filter residues (stick and line modes). Only two diagonal pore domains are shown for clarity. (**B**) Representative current traces from BK channels formed by single $\alpha_M$ and concatenated $\alpha_{M(dual)}$ and $\alpha_{M(quad)}$ constructs containing subunit-specific L312A mutations. Depolarizations from −80 mV were applied in 20 mV steps. Mutated subunits are indicated by filled circles and WT subunits by empty circles. (**C**) Voltage dependence of activation for channels formed by the indicated L312A mutant $\alpha_M$ and $\alpha_{M(dual)}$ constructs co-expressed with $\alpha^{\Delta M}$. Dashed lines show G-V curves of the corresponding non-mutated channels for comparison. (**D**) Voltage dependence of activation for channels formed by the $\alpha_{M(quad)}$ constructs with different numbers of L312A mutations. A plot of the $V_{1/2}$ vs. the number of mutated subunits is shown. (**E**) Plot of tail current decay rates (–120 mV) vs. number of L312A mutations, from $\alpha_{M(dual)}$ and $\alpha_{M(quad)}$ constructs. (**F**) Voltage dependence of activation for channels formed by co-expression of WT and L312A-mutant intact BKα subunits (n=6), or non-mutated and fully mutated $\alpha_{M(quad)}$ constructs co-expressed with $\alpha^{\Delta M}$. (**G**) Representative current traces from channels formed by co-expressing the non-mutated and fully L312A mutated $\alpha_{M(quad)}$ constructs (co-expressed with $\alpha^{\Delta M}$). Enlarged and fitted tail currents are shown below. Electrophysiological recordings were performed under Ca$^{2+}$-free conditions and repeated n=4–10 as indicated in *Table 1*. Error bars represent ± SEM.

two distinct exponential decay components differing by ~10-fold in rate, corresponding to populations of all-WT and all-mutated channels (*Figure 2G*). These findings support that the concatenated $\alpha_M$ repeats in the $\alpha_{M(dual)}$ and $\alpha_{M(quad)}$ constructs function as a whole unit in channel assembly, enabling stoichiometrically defined investigation of channel gating and regulation.

## V288A triggers selectivity filter inactivation through an all-subunit mechanism

The selectivity filter in $K^+$ channels is directly involved in C-type inactivation. Given its potential role in BK channel activation gating (*Piskorowski and Aldrich, 2006*; *Yan et al., 2016*), we examined the subunit stoichiometric effects of structural perturbations within the selectivity filter on BK channel gating. We found that the V288A mutation, located within the $K^+$-selective signature sequence ($^{286}$STVGYGD$^{292}$) (*Figure 3A*), produced profound effects on BK channel gating. We previously reported that mutations near the selectivity filter, for example, in the P-helix (Y279) or at the extracellular side of the filter (Y294), can induce an atypical closed-state-coupled C-type inactivation in BK channels under low extracellular $K^+$ conditions (*Yan et al., 2016*). Interestingly, V288A, even without reduced extracellular $K^+$, induced a similar slow inactivation process, causing a gradual decrease in the availability of activatable channels under conditions (e.g. negative voltages) that promote channel closure (*Figure 4A*). Consequently, compared to WT channels, V288A mutant channel currents developed very slowly in response to depolarization (*Figure 4B*). However, following a long pre-depolarization, the mutant channels fully recovered from the inactivated state and behaved similarly to WT channels in the kinetics and voltage dependence of activation (*Figure 4C*).

Using the concatenated $BK\alpha_M$ constructs, we investigated the subunit stoichiometry of V288A-induced gating effects by introducing the mutation into different numbers of $BK\alpha$ subunits within a single channel. With the $BK\alpha_{M(dual)}$ construct, we generated $\alpha_M{}^{V288A}\alpha_M{}^{WT}$ and $\alpha_M{}^{V288A}\alpha_M{}^{V288A}$ constructs, harboring the mutation on half or all subunits, respectively. We introduced one ($\alpha_M{}^{V288A}\alpha_M{}^{WT}\alpha_M{}^{WT}\alpha_M{}^{WT}$), two ($\alpha_M{}^{V288A}\alpha_M{}^{V288A}\alpha_M{}^{WT}\alpha_M{}^{WT}$), three ($\alpha_M{}^{V288A}\alpha_M{}^{V288A}\alpha_M{}^{WT}\alpha_M{}^{V288A}$), or four ($\alpha_M{}^{V288A}\alpha_M{}^{V288A}\alpha_M{}^{V288A}\alpha_M{}^{V288A}$) mutations on the $BK\alpha_{M(quad)}$ construct. Interestingly, unlike the one-subunit-sufficient effect of LRRC26 or the stoichiometrically incremental effects of the L312A mutation, V288A exerted an "all-subunit-required" modulatory effect: the V288A-induced changes occurred only when all four $BK\alpha$ subunits in a channel were mutated. Channels partially mutated, originated from $\alpha_M{}^{V288A}\alpha_M{}^{WT}$, $\alpha_M{}^{V288A}\alpha_M{}^{WT}\alpha_M{}^{WT}\alpha_M{}^{WT}$, $\alpha_M{}^{V288A}\alpha_M{}^{V288A}\alpha_M{}^{WT}\alpha_M{}^{WT}$, or $\alpha_M{}^{V288A}\alpha_M{}^{V288A}\alpha_M{}^{WT}\alpha_M{}^{V288A}$, showed no significant differences from WT channels in the time course (*Figure 4D and E*) or voltage-dependence (*Figure 4F*) of depolarization-induced currents. In contrast, channels with all $\alpha_M$ repeats mutated ($\alpha_M{}^{V288A}\alpha_M{}^{V288A}$ and $\alpha_M{}^{V288A}\alpha_M{}^{V288A}\alpha_M{}^{V288A}\alpha_M{}^{V288A}$) displayed markedly slowed current development, approximately 100-fold slower than WT (*Figure 4D and E*). These fully mutated channels also showed an apparently higher $V_{1/2}$ in their G-V relationships (*Figure 4F*). However, since the V288A mutation has no significant effect on the channel's normal activation gating (*Figure 4C and F*), this apparent shift in G-V curves likely resulted from incomplete recovery of inactivated channels during 50ms depolarization phase in the voltage protocol used. Together, these results demonstrate that V288A induces an "all-subunit-required" inactivation process at the selectivity filter, while the activation/deactivation processes of normal channel gating remain largely unchanged.

## Discussion

Concatenated constructs have been extensively used to study ion channel subunit stoichiometry (*Hurst et al., 1995*; *Fahlke et al., 1998*; *Minier and Sigel, 2004*; *White, 2006*; *Ogielska et al., 1995*; *Zandany et al., 2008*; *Wu et al., 2014*), enabling a deeper understanding of the mechanisms underlying channel gating and regulation by homo- or heteromeric subunits, voltage, ligands, and mutations. However, the BK channel is one of the few channels whose principal subunits have their N-termini located on the extracellular side. This unique membrane topology prevents the direct application of the traditional N-to-C-terminal concatenation method for generating the concatenated constructs. In this study, we employed a strategy that involved splitting and fusing $BK\alpha$ subunits into two modular constructs that reconstitute functional BK channels. We validated the functionality of these concatenated constructs by demonstrating that the resulting channels closely resemble intact BK channels in their voltage and $Ca^{2+}$ dependence of activation. Furthermore, we confirmed that each repeat with the constructs contributes similarly to channel function, as evidenced by the

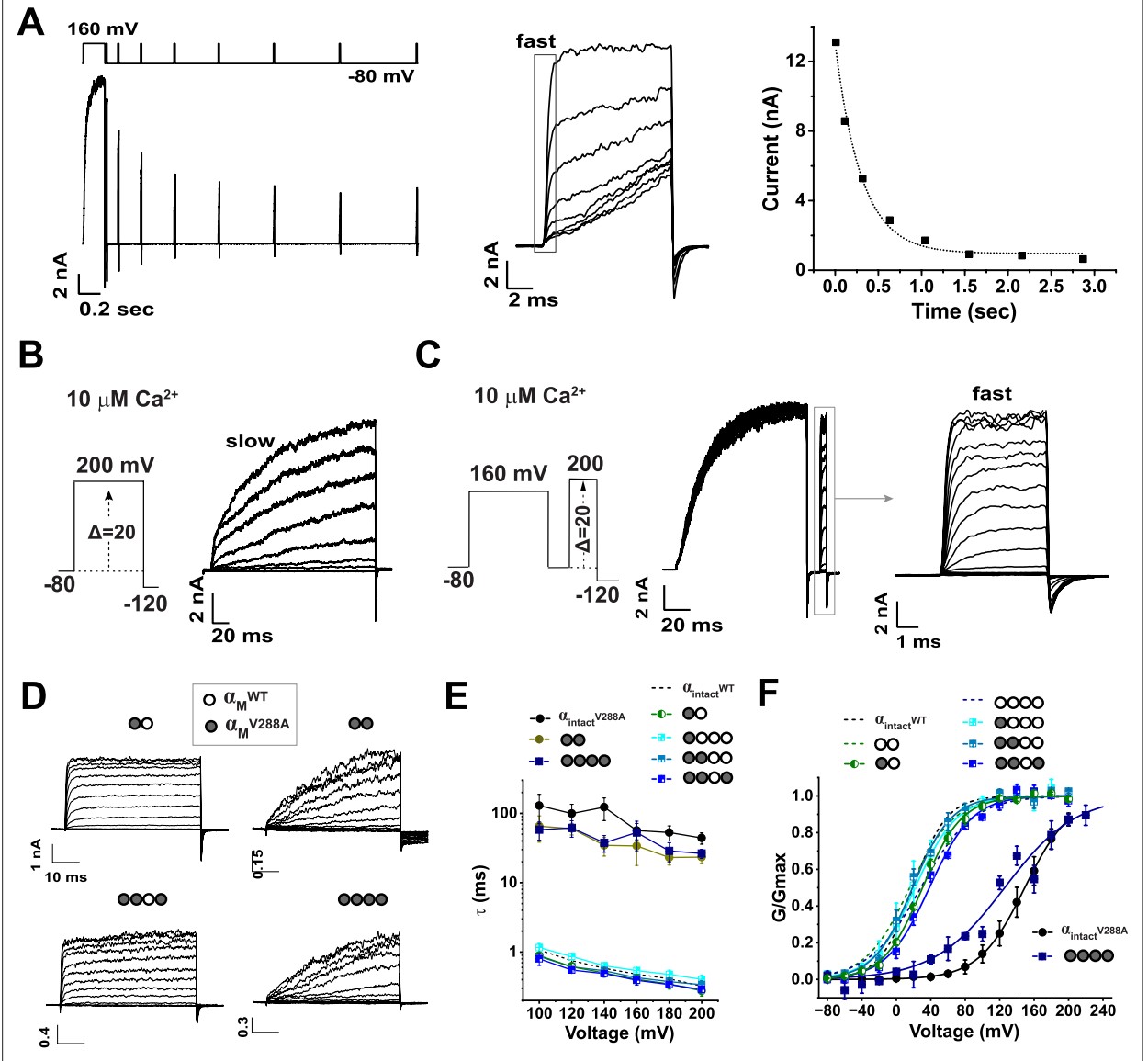

**Figure 4.** V288A-induced selectivity filter inactivation requires mutation of all subunits. (**A**) Time-dependent inactivation of V288A-mutant BKα (intact) channels at –80 mV after a prolonged depolarization (160 mV for 200ms). Inactivation was assayed by monitoring the reduction in fast-activating currents (shown and compared in the middle) elicited by brief depolarization (160 mV for 10ms) after a prior –80 mV holding time of 10ms, 0.1 s, 0.2 s, 0.3 s, 0.4 s, 0.5 s, 0.6 s, and 0.7 s. The amplitudes of fast-activating currents are compared (middle) and plotted against time (right). (**B**) Representative current traces of V288A mutant BKα (intact) channels showing slowly developing depolarization-induced currents. (**C**) V288A mutant channel exhibited normal activation gating following recovery (160 mV for 100ms) from inactivation, as indicated by currents elicited by brief depolarization to different voltages after brief repolarization. (**D**) Representative current traces from channels formed by concatenated $\alpha_{M(dual)}$ and $\alpha_{M(quad)}$ constructs with subunit-specific V288A mutations (co-expressed with $\alpha^{\Delta M}$). Mutant and WT subunits are indicated as filled and empty circles, respectively. (**E**) Depolarization-induced current development rates for channels formed by non-mutated and V288A-mutant BKα (intact) and concatenated $\alpha_{M(dual)}$ and $\alpha_{M(quad)}$ constructs. Electrophysiological repeats: n=8 for $\alpha_{(intact)}^{WT}$, 5 for $\alpha_{(intact)}^{V288A}$, 4 for $\alpha_M^{V288A}{}_M^{WT}$, 4 for $\alpha_M^{V288A}{}_M^{V288A}$, 4 for $\alpha_M^{V288A}{}_M^{WT}{}_M^{WT}{}_M^{WT}$, 4 for $\alpha_M^{V288A}{}_M^{V288A}{}_M^{WT}{}_M^{WT}$, 3 for $\alpha_M^{V288A}{}_M^{V288A}{}_M^{WT}{}_M^{V288A}$, and 4 for $\alpha_M^{V288A}{}_M^{V288A}{}_M^{V288A}{}_M^{V288A}$. (**F**) Voltage dependence of depolarization-induced currents for BK channels formed by non-mutated and V288A-mutant BKα (intact) and concatenated $\alpha_{M(dual)}$ and $\alpha_{M(quad)}$ constructs. Electrophysiological repeats n=3–5 as indicated in **Table 1**. All recordings were performed using symmetric K⁺ (140 mM) solutions with 10 µM intracellular Ca²⁺. Error bars represent ± SEM.

stoichiometrically incremental effect of the L312A mutation on voltage gating. Previous studies have reported that, in some cases, subunits from different quadruple-repeat concatemers can assemble aberrantly, leading to the formation of mixed channels with altered properties (*McCormack et al., 1992*; *Sack et al., 2008*). The large functional effect of the L312A mutation on BK channel gating allowed us to distinguish between properly assembled channels and potential inter-subunit crossover

products. Our results showed that the concatenated tandem constructs assemble predominantly as intended, that is each channel is formed by two dual-repeat constructs or a single quadruple-repeat construct, as evidenced by the absence of significant heterogeneity in channel gating properties. Using these well-defined concatenated constructs, we identified three distinct types of subunit stoichiometry in BK channel gating or modulation. These rule out the possibility that the observed stoichiometric effects are artifacts arising from the construct design or assembly defects. Thus, we have demonstrated that our engineered concatenated BKα constructs serve as effective molecular tools for probing the subunit stoichiometry in BK channel gating and regulation.

Most auxiliary proteins of the $K^+$ channels exhibit stoichiometrically incremental effects on channel modulation. However, the auxiliary γ1 subunit displays an unusual binary 'all-or-none' modulatory effect on BK channels (*Chen et al., 2022*; *Gonzalez-Perez et al., 2014*), despite being able to bind to BKα subunits at a 1:1 molecular ratio (*Gonzalez-Perez et al., 2018*; *Noda et al., 2020*; *Carrasquel-Ursulaez et al., 2018*). In recently reported cryo-EM structures of channel complexes, the four γ1 subunits display an apparent four-fold symmetry in TM domain interactions with their LRR domains tetramerized on the extracellular side (*Kallure et al., 2023*; *Yamanouchi et al., 2023*; *Redhardt et al., 2024*). These structural features raise the possibility that the four γ1 subunits might act collectively in BK channel modulation (*Kallure et al., 2023*), which appears to contradict the 'one-subunit-sufficient' mechanism previously inferred from single channel gating properties of BK channels modulated by a β2-γ1 chimeric construct (*Gonzalez-Perez et al., 2018*). With the concatenated modular BKα constructs developed in this study, we were able to directly control the subunit stoichiometry of γ1 subunits relative to BKα subunits. The intracellular location of the N-terminus in the concatenated BKα constructs enabled us to fuse the γ1 subunit to the N-terminal side of the $α_M$ module. With γ1-fused concatenated $α_M$ dual- and quadruple-repeat constructs, we provide direct evidence that one and two copies of the γ1 subunits per BKα tetramer are sufficient to produce the full modulatory effect of the γ1 subunit on BK channel gating. Thus, our findings unequivocally confirm the 'one-subunit-sufficient' mechanism of the γ1 subunit in BK channel modulation by using the full length γ1 subunit, stoichiometrically defined BKα/γ1 channel complexes, and macroscopic currents from large channel populations. This result, combined with the structural observation of symmetrical binding of γ1 to all four voltage-sensor domains, raises the intriguing possibility that γ1 modulation may occur through asymmetric allosteric coupling, despite symmetric structural binding—a phenomenon warranting further mechanistic investigation.

Through conformational linkage to the voltage- and $Ca^{2+}$-sensors, the movement of the lower half of the pore-lining S6 helix ultimately controls BK channel pore gating. According to the Horrigan-Aldrich gating model (*Horrigan and Aldrich, 2002*), BK channel activation involves a rate-limiting process of pore-gate opening, regulated by four independent and identical voltage- and $Ca^{2+}$-sensors. However, the subunit stoichiometry underlying the S6 movement-induced pore-gate opening in BK channels remains unclear and warrants investigation. The deep pore residue L312 on S6 is unique in that it lies adjacent to the double-glycine gating hinge residues (G310 and G311), is positioned immediately below the selectivity filter (*Figure 3A*), and appears to be the most mutation-sensitive residue affecting channel activation gating, as most of its substitution mutations result in constitutively open channels (*Chen et al., 2014*). Thus, L312 appears to represent a structural endpoint for the voltage- and $Ca^{2+}$-induced conformational changes in S6 and plays an essential role in stabilizing the channel's closed state. In classic Shaker $K^+$ channels, a mutation of the neighboring glycine hinge residue (G466 in Shaker, corresponding to G311 in BKα) in concatenated tetrameric constructs was reported to display a concerted, 'one-subunit-sufficient' effect, where all mutant subunit combinations produced similar effects on channel gating (*Zandany et al., 2008*), a phenomenon reminiscent of the modulatory behavior of the LRRC26 (γ1) subunit on BK channel gating. In contrast, our current study reveals a stoichiometrically graded (independent) effect of the L312A mutation on BK channel voltage gating. Similarly, single-channel recordings showed that a mutation to arginine at the neighboring G311 residue also produced an additive effect on the BK channel voltage-gating proportional to the number of mutated subunits (*Geng et al., 2023*). Together, the observed stoichiometric independence of mutational effects at the L312 and G311 residues is consistent with the modeled independence of individual voltage and $Ca^{2+}$ sensors in channel activation (*Horrigan and Aldrich, 2002*; *Geng et al., 2023*), and likely reflects a fundamental difference in the location and mechanism of the activation gate. Currently, the fundamental question of pore gate location in BK channels remains unsettled.

In most K$^+$ channels, a hydrophobic 'bundle-crossing' gate near the intracellular end of the pore controls channel activation (*Aryal et al., 2015*). However, BK channels appear to lack such a gate, as the pore remains structurally wide open in the presumed closed-state (Ca$^{2+}$-free) structures (*Tao and MacKinnon, 2019*; *Hite et al., 2017*) and is readily accessible to large intracellular blockers (*Wilkens and Aldrich, 2006*; *Thompson and Begenisich, 2012*) or cysteine-modifying reagents (*Zhou et al., 2011*) even when the channel is closed. Thus, in Shaker and related Kv channels, the classical activation gate resides within the lower S6 helices, where concerted subunit movements are required for highly cooperative transitions from closed to open states. In contrast, the graded, additive effects of deep pore mutations in BK channels provide evidence that no such concerted gating structure exists within the lower S6 or at least not within the deep pore region.

The selectivity filter, known to govern C-type inactivation in many channels (*Yellen, 1998*; *Kukuljan et al., 1995*; *Cuello et al., 2010*), may also serve as activation gate in some, such as the ligand-gated CNG channels (*Contreras et al., 2008*). Accordingly, direct involvement of the selectivity filter in BK channel activation gating has long been speculated (*Minier and Sigel, 2004*; *Piskorowski and Aldrich, 2006*; *Wilkens and Aldrich, 2006*), but compelling experimental validation remains lacking. While C-type inactivation doesn't normally occur in BK channels, we previously found that it can be induced by mutations near the selectivity filter in combination with low extracellular K$^+$ (*Yan et al., 2016*). Interestingly, the induced C-type inactivation in BK channels is closed-state coupled (*Yan et al., 2016*), opposite to the open-state coupled C-type inactivation commonly observed in other channels. In this study, we report that the V288A mutation, located within the K$^+$-selective signature sequence, also induces pronounced inactivation even under normal extracellular K$^+$ conditions. Our analysis of the subunit stoichiometric effects of V288A using concatenated BKα dual- and quadruple-repeat constructs clearly showed that modifications in all four subunits are required to elicit the mutation-induced inactivation, whereas all other mutant subunit combinations produced minimal effects on BK channel gating. It is noteworthy that the stoichiometric effects of mutations on the rate of C-inactivation vary depending on the location of the mutated residue in other voltage-gated K$^+$ channels. Mutations located more peripherally from the selectivity filter signature motif often produced either a non-linear, graded effect consistent with a cooperative inter-subunit interaction (e.g. A463V, A449K, and W434F in Shaker channels, or S631A and T618A in hERG1; *Ogielska et al., 1995*; *Wu et al., 2014*; *Yang et al., 1997*), or a linearly additive effect consistent with inter-subunit independence (e.g. M645C in hERG1; *Wu et al., 2014*). In contrast, mutations near or within the selectivity filter signature motif affected C-type inactivation in an all-or-none manner, with a single subunit mutation, such as S620T or G628C in hERG1, being sufficient to eliminate or attenuate inactivation to the same extent as observed in mutant homotetramers (*Wu et al., 2014*). This effect closely mirrors our observations with V288A in BK channels, where mutations in all four subunits are required to induce and maintain inactivation, supporting our interpretation that V288A induces a form of C-type inactivation, similar to what we previously observed with mutations at the extracellular mouth and the P-helix (*Yan et al., 2016*). Given its critical location, the V288A mutation may alter BK channel ion selectivity, as an alanine substitution at the equivalent position in Kv4.3 has been shown to reduce K$^+$ selectivity (*Strutz-Seebohm et al., 2013*). Whether V288A induces both C-type inactivation and changes in ion selectivity through a similar subunit stoichiometry remains to be determined.

The contrasting behaviors of L312A and V288A mutations in concatenated BK α$_M$ constructs reveal distinct subunit stoichiometry requirements between the deep pore activation gating and selectivity filter inactivation gating. A typical K$^+$ channel activation gate requires all four subunits to undergo a concerted conformational change to ensure an all-or-none transition between a fully conductive and non-conductive pore (*Zagotta et al., 1994*; *Schoppa and Sigworth, 1998*), a hallmark of classic voltage-gated K$^+$ channels that allows rapid and precise control of membrane excitability. Such tight inter-subunit all-or-none cooperativity observed at the BK channel selectivity filter makes it a plausible candidate for serving as the activation gate, a property not yet demonstrated for the lower S6 segment. It is possible that the selectivity filter functions as the physical gate for two gating processes, activation and inactivation, that can occur on different timescales and through distinct structural mechanisms. Previously, we found that the voltage- and Ca$^{2+}$-dependence of inactivation and activating gating were well correlated in BK channels (*Yan et al., 2016*), suggesting a shared or coupled energetic pathway. L312 lies in close proximity to and physically interacts with the selectivity filter, providing a possible structural link for transmitting energy from S6 movement to the selectivity filter

during activation gating. Further investigation of inactivation and its relationship to activation will likely help elucidate the role of the selectivity filter in BK channel activation gating.

In conclusion, our study employed an innovative strategy to generate concatenated subunit constructs and investigate the subunit stoichiometry and modulation of BK channels. The development of these constructs enabled detailed exploration of the intricate gating and regulatory mechanisms of BK channels in a stoichiometrically subunit-specific manner. Using these concatenated constructs, we identified three distinct types of subunit stoichiometry in BK channel modulation: an additive (independent) type and two contrasting all-or-none types, namely, 'one-subunit-sufficient' and 'all-subunit-required'. These represent divergent stoichiometric modes of gating control by the pore, LRRC26 (γ1), and selectivity filter, respectively. This study offers new molecular tools and advances our understanding of subunit stoichiometry in BK channel gating and modulation.

## Materials and methods
### Generation of concatenated tandem BKα repeat constructs and expression of BK channels

We first generated a pcDNA6-based plasmid, pcDNA6-myc-BKα-V5-His, carrying *KCNMA1* cDNA (GenBank: U11058). This plasmid expresses the full-length (1113 amino acids) human BKα (GenBank: AAB65837) with an N-terminal Myc tag and C-terminal V5 and 6×His tags, serving as the template for further plasmid constructions. To express BKα's main region (residues 44–651) as a protein module, we created pcDNA6-myc-$\alpha_M$-V5-His by deleting the nucleotide sequences encoding N-terminal residues 1–43 (extracellular N-terminus and S0 TM segment) and C-terminal residues 652–1113 (RCK2 domain and C-terminal tail). Next, we constructed a complementary plasmid, pcDNA6-myc-BKα$^{\Delta M}$-GFP-V5-His, by replacing residues 94–651 of BKα with a flexible peptide linker (SSGGGGSGGGS-GGAR) and tagging monomeric enhanced GFP to the C-terminus. This complementary plasmid enables functional channel formation when co-expressed with a plasmid encoding a single $\alpha_M$ or concatenated $\alpha_M$ repeats. To construct structurally stable plasmids expressing concatenated $\alpha_M$ repeats, we synthesized three codon-optimized DNA sequences encoding the same $\alpha_M$ module, each differing by ~25% in nucleotide sequence from each other and the original *KCNMA1* cDNA. Using these synthesized sequences, we constructed the dual-repeat expressing plasmid, pcDNA6-$\alpha_{M1}\alpha_{M2}$-V5-His, in which $\alpha_{M1}$ (residues 43–649) is preceded by a short initiation sequence (MGS) and linked to $\alpha_{M2}$ (also residues 43–649) via a flexible linker (GGGGSGSAG). A NotI restriction site with a peptide spacer (GGGKPIPNAAA) was inserted between $\alpha_{M2}$ and the V5 tag. We also generated a second dual-repeat expressing plasmid, the pcDNA6-$\alpha_{M3}\alpha_{M4}$-V5-His, in which $\alpha_{M3}$ (residues 44–649) is preceded by an N-terminal sequence (MGAAAA) containing a NotI site and linked $\alpha_{M4}$ (residues 43–649) via a flexible linker (GGGSAAGSG). As the two dual-repeat constructs produce highly similar proteins and exhibit no difference in electrophysiological properties, both are referred to as BKα$_{(dual)}$ or $\alpha_{MM}$. To generate a quadruple-repeat expressing plasmid, pcDNA6-$\alpha_{M1}\alpha_{M2}\alpha_{M3}\alpha_{M4}$-V5-His, we subcloned the $\alpha_{M3}\alpha_{M4}$ dual-module fragment from pcDNA6-$\alpha_{M3}\alpha_{M4}$-V5-His into pcDNA6-$\alpha_{M1}\alpha_{M2}$-V5-His using NotI and AgeI (located between the V5 and 6×His tags). The expressed protein is referred to as BKα$_{(quad)}$ or $\alpha_{MMMM}$. For γ1 (LRRC26) fusion constructs, we generated pcDNA6-BKγ1$\alpha_{M1}\alpha_{M2}$-V5-His and by inserting the γ1 (LRRC26) sequence at the N-terminus of $\alpha_{M1}\alpha_{M2}$ (pcDNA6-$\alpha_{M1}\alpha_{M2}$-V5-His) via a 16-residue flexible linker (SSGSGSESKSTGGSGS). The expressed fusion protein is designated as BKγ1$\alpha_{MM}$ or BKγ1$\alpha_{M(dual)}$. To express the BKγ1$\alpha_{MMMM}$ (also referred to as BKγ1$\alpha_{M(quad)}$) fusion construct, we created pcDNA6-BKγ1$\alpha_{M1}\alpha_{M2}\alpha_{M3}\alpha_{M4}$-V5-His by subcloning $\alpha_{M3}\alpha_{M4}$ from pcDNA6-$\alpha_{M3}\alpha_{M4}$-V5-His into pcDNA6-BKγ1$\alpha_{M1}\alpha_{M2}$-V5-His using NotI and AgeI. For DNA manipulation and amplification, we used the Long Fragment DNA Ligation Kit (TaKaRa) and CopyCutter competent *E. coli* (Lucigen). Site-directed mutagenesis was performed using the QuickChange kit (Stratagene). Mutations for the quadruple-repeat construct were first introduced into either the $\alpha_{M1}\alpha_{M2}$ or $\alpha_{M3}\alpha_{M4}$ dual-repeat construct, followed by fusion of $\alpha_{M1}\alpha_{M2}$ and $\alpha_{M3}\alpha_{M4}$ as described above. HEK293 cells (CRL-1573 from ATCC; authenticated with STR profiling and tested negative for mycoplasma contamination) were cultured in DMEM supplemented with 10% fetal bovine serum at 5% $CO_2$. Cells were transfected with plasmids using PEI 'MAX' (Polysciences Inc) and subjected to electrophysiological assays 16–72 hr post-transfection.

## Electrophysiology

BK channel currents were recorded from excised inside-out patches of HEK293 cells using patch-clamp recording techniques as described previously (*Chen et al., 2023*). Both intracellular and extracellular (pipette) solutions contained 136 mM KMeSO$_3$, 4 mM KCl, and 20 mM HEPES (pH 7.20). The extracellular solution was supplemented with 2 mM MgCl$_2$, while the intracellular solution contained 5 mM HEDTA either with or without Ca$^{2+}$ to achieve 10 µM Ca$^{2+}$ or Ca$^{2+}$free. Recording pipette electrodes were pulled from borosilicate filamented glass tubes (Cat #: BF150-110-10, Sutter Instrument) with a P-1000 micropipette puller (Sutter Instrument), and polished by heat with an MF-830 microforge (Narishige) to a resistance of 1–2 MΩ. Data were acquired using PatchMaster (HEKA) with an Axopatch 200B amplifier (Molecular Devices) and ITC-18 digitizer (InstruTECH) or with an EPC-10 amplifier (HEKA). Data were sampled at 20 µs and filtered at 2 kHz (Axopatch 200B) with the amplifiers' 4-pole Bessel filter or at 2.9 kHz (EPC-10). Time interval between voltage protocol sweeps was 2 s. Capacitive and leak currents were subtracted using a P/4 protocol at holding potentials of –120 mV or –150 mV (for γ1 or 10 µM Ca$^{2+}$ conditions). Steady-state activation, expressed as normalized conductance (G/Gmax) versus voltage (G-V), was calculated from the tail current amplitudes (at –120 mV) and fitted using a single-Boltzmann function $G/Gmax = 1/(1+e^{-ZF(V-VH)/RT})$ or a double-Boltzmann function $G/Gmax = Pa/(1+e^{-ZaF(V-VHa)/RT}) + (1 - Pa)/(1+e^{-ZbF(V-VHb)/RT})$ where V, VH, Z, F, R, T, Pa, a, and b denote voltage, $V_{1/2}$, gating charge (z), Faraday constant, gas constant, Kelvin temperature, component portion (0–1), and component identity (a or b), respectively. Values are reported as means ± SEM.

## Immunoblotting

Proteins were enriched by immunoprecipitation as previously described (*Chen et al., 2022*) and immunoblotted after SDS-PAGE. Briefly, proteins were solubilized from cells in 2% Dodecyl-beta-D-maltoside (DDM) in TBS buffer (50 mM Tris, 150 mM NaCl, pH 7.6). Lysates were incubated with mouse anti-V5 monoclonal antibody agarose gel (Cat# A7345, Millipore Sigma) at 4 °C for 2 hr. After three 10 min washes with 2% DDM-containing TBS, bound proteins were eluted with 4% SDS. Protease inhibitor cocktail (Roche) was used throughout the procedure. Eluted proteins were separated by 4% to 20% gradient SDS-PAGE and transferred to PVDF membranes. Immunoblotting was performed with mouse anti-V5 monoclonal antibody (Cat# R96125, Invitrogen) at 1:10,000 dilution.

## Materials availability

Newly generated materials from this study will be made available upon reasonable request, without licensing or patent-related restrictions.

## Acknowledgements

This work was supported by National Institutes of Health grants NS078152 (JY) and GM127332 (JY).

## Additional information

### Funding

| Funder | Grant reference number | Author |
| --- | --- | --- |
| National Institute of Neurological Disorders and Stroke | NS078152 | Jiusheng Yan |
| National Institute of General Medical Sciences | GM127332 | Jiusheng Yan |

The funders had no role in study design, data collection and interpretation, or the decision to submit the work for publication.

## Author contributions
Guanxing Chen, Data curation, Formal analysis, Investigation, Methodology, Writing – original draft; Qin Li, Kunal Shah, Data curation; Jiusheng Yan, Conceptualization, Data curation, Formal analysis, Supervision, Funding acquisition, Investigation, Methodology, Writing – original draft, Project administration, Writing – review and editing

## Author ORCIDs
Kunal Shah  https://orcid.org/0000-0002-4721-1169
Jiusheng Yan  https://orcid.org/0000-0002-6633-1799

Reviewer #1 (Public review): https://doi.org/10.7554/eLife.107681.3.sa1
Reviewer #2 (Public review): https://doi.org/10.7554/eLife.107681.3.sa2
Author response https://doi.org/10.7554/eLife.107681.3.sa3

---

## Additional files

### Supplementary files
MDAR checklist

### Data availability
All data relevant to this work is presented in the manuscript and supporting files.

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
