## [Editor Report · eLife Assessment]

In this **important** contribution, Yan and colleagues describe a powerful and **compelling** strategy to generate concatamers of the BK channel and their fusion constructs with the auxiliary gamma subunits, which allows exploring contributions of individual subunits of the tetrameric channel to its gating and the study of heteromeric channel complexes of defined composition. Distinct examples are presented, which illustrate great diversity in the stoichiometric control of BK channel gating, depending on the site and nature of molecular perturbations. The molecular approaches could be extended to other membrane proteins whose N and C termini face opposite sides of the membrane.

---

## [Referee Report · Reviewer #1 (Public review)]

Summary:

BK channels are widely distributed and involved in many physiological functions. They have also proven a highly useful tool for studying general allosteric mechanisms for gating and modulation by auxiliary subunits. Tetrameric BK channels are assembled from four separate alpha subunits which would be identical for homozygous alleles and of potentially five different combinations for heterozygous alleles Geng et al . (2023), (https://doi.org/10.1085/jgp.202213302). Construction of BK channels with concatenated subunits in order to strictly control heteromeric subunit composition had not yet been used because the N-terminus in BK channels is extracellular whereas the C-terminus is intracellular. In this new work, Chen, Li, and Yan devise clever methods to construct and assemble BK channels of known subunit composition, as well as to fix the number of γ1 axillary subunits per channel. With their novel molecular approaches, Chen, Li and Yan report that a single γ1 axillary subunit is sufficient to fully modulate a BK channel, that the deep conducting pore mutation L312A exhibited a graded effect on gating with each addition mutated subunit replacing a WT subunit in the channel adding an additional incremental left shift in activation, and that the V288A mutation at the selectivity filter must be present on all four alpha subunits in order to induce channel inactivation. Chen, Li, and Yan have been successful in introducing new molecular tools to generate BK channels of known stoichiometry and subunit composition. They validate their methods and provide three different examples of stoichiometric modulation by LRRC26, the selectivity filter, and the pore.

Strengths:

Powerful new molecular tools for study of channel gating are developed and validated in the study.

Weaknesses:

One example each of auxiliary, deep pore, and selectivity filter allosteric actions are presented, but this is sufficient for the purposes of the paper to establish their methods and present specific examples of applicability.

---

## [Referee Report · Reviewer #2 (Public review)]

Summary:

This manuscript describes novel BK channel concatemers as a tool to study the stoichiometry of gamma subunit and mutations in modulation of the channel. Taking the advantage of modular design of BK channel alpha subunit the authors connected S1-S6/1st RCK as two- and four-subunit concatemers and coexpressed with S0-RCK2 to form normal function channels. These concatemers avoided the difficulty that the extracellular N-terminus of S0 was unable to connect with the cytosolic C-terminus of the alpha or gamma subunit, allowing a single gamma subunit to be connected to the concatemers. The concatemers also helped reveal the required stoichiometry of mutant BK subunits in modulating channel function. These include L312A in the deep pore region that altered channel function additively with each additional subunit harboring the mutation, and V288A at the selectivity filter that altered channel function cooperatively only when all four subunits being mutated. These results demonstrate that the concatemers are robust and effective in studying BK channel function and molecular mechanisms related to stoichiometry. The different requirement of the gamma subunit and the mutations stoichiometry for altering channel function is interesting, revealing fundamental mechanisms of how different motifs of the channel protein control function.

Strengths:

The manuscript presents well designed experiments with high quality data, which convincingly demonstrate the BK channel concatemers and their utility. The results are clearly written.

Weaknesses:

This reviewer did not identify any major concerns with the manuscript.

Editors' note: We thank you for addressing some of the concerns, adding clarifications and more complete discussions, including further details about experimental protocols. The revised version is significantly improved. Some concerns linger that the biophysical/structural mechanisms underlying the observed phenotypes remain unclear and in some ways are phenomenological. However, the current study is more about the methodology and the mechanisms underlying the stoichiometry dependent effects are perhaps left for a separate study, with more detailed exploration. Congratulations for the excellent work.

---

## [Author Response]

The following is the authors’ response to the original reviews.

**Public Reviews:**

**Reviewer #1 (Public review):**
Summary:BK channels are widely distributed and involved in many physiological functions. They have also proven a highly useful tool for studying general allosteric mechanisms for gating and modulation by auxiliary subunits. Tetrameric BK channels are assembled from four separate alpha subunits, which would be identical for homozygous alleles and potentially of five different combinations for heterozygous alleles (Geng et al., 2023, https://doi.org/10.1085/jgp.202213302). Construction of BK channels with concatenated subunits in order to strictly control heteromeric subunit composition had not yet been used because the N-terminus in BK channels is extracellular, whereas the C-terminus is intracellular. In this new work, Chen, Li, and Yan devise clever methods to construct and assemble BK channels of known subunit composition, as well as to fix the number of γ1 axillary subunits per channel. With their novel molecular approaches, Chen, Li and Yan report that a single γ1 axillary subunit is sufficient to fully modulate a BK channel, that the deep conducting pore mutation L312A exhibited a graded effect on gating with each addition mutated subunit replacing a WT subunit in the channel adding an additional incremental left shift in activation, and that the V288A mutation at the selectivity filter must be present on all four alpha subunits in order to induce channel inactivation. Chen, Li, and Yan have been successful in introducing new molecular tools to generate BK channels of known stoichiometry and subunit composition. They validate their methods and provide three examples of their use with useful observations.Strengths:Powerful new molecular tools for the study of channel gating have been developed and validated in the study.Weaknesses:(1) One example each of auxiliary, deep pore, and selectivity filter allosteric actions is presented, but this is sufficient for the purposes of the paper to establish their methods and present specific examples of applicability.

We sincerely thank Reviewer #1 for the thoughtful and supportive evaluation of our work. We greatly appreciate the reviewer’s clear summary of the study and the recognition of the novelty and utility of our molecular concatemer strategy for controlling BK channel subunit composition and stoichiometry.

We also appreciate the reviewer’s positive assessment that the three examples (auxiliary subunit modulation, deep pore mutation, and selectivity filter mutation) are sufficient to establish the method and demonstrate its applicability. We are encouraged that the reviewer found the new molecular tools to be powerful and well validated.

We have no further changes to make in response to this review, but we are grateful for the reviewer’s constructive and encouraging comments.

**Reviewer #2 (Public review):**
Summary:This manuscript describes novel BK channel concatemers as a tool to study the stoichiometry of the gamma subunit and mutations in the modulation of the channel. Taking advantage of the modular design of the BK channel alpha subunit, the authors connected S1-S6/1st RCK as two- and four-subunit concatemers and coexpressed with S0-RCK2 to form normal function channels. These concatemers avoided the difficulty that the extracellular N-terminus of S0 was unable to connect with the cytosolic C-terminus of the gamma subunit, allowing a single gamma subunit to be connected to the concatemers. The concatemers also helped reveal the required stoichiometry of mutant BK subunits in modulating channel function. These include L312A in the deep pore region that altered channel function additively with each additional subunit harboring the mutation, and V288A at the selectivity filter that altered channel function cooperatively only when all four subunits were mutated. These results demonstrate that the concatemers are robust and effective in studying BK channel function and molecular mechanisms related to stoichiometry. The different requirement of the gamma subunit and the mutations stoichiometry for altering channel function is interesting, which may relate to the fundamental mechanism of how different motifs of the channel protein control function.Strengths:The manuscript presents well-designed experiments with high-quality data, which convincingly demonstrate the BK channel concatemers and their utility. The results are clearly presented.Weaknesses:This reviewer did not identify any major concerns with the manuscript.

We sincerely thank Reviewer #2 for the careful reading of our manuscript and for the highly positive and supportive comments. We appreciate the reviewer’s detailed summary of our concatemer design strategy and its use in studying gamma subunit stoichiometry and mutation-dependent modulation of BK channel function.

We are especially grateful for the reviewer’s recognition that the experiments are well designed, the data are of high quality, and the results demonstrate the robustness and utility of the concatemer approach. We also appreciate the reviewer’s thoughtful note on the mechanistic implications of the distinct stoichiometric requirements observed for the gamma subunit, L312A, and V288A.

We are pleased that the reviewer identified no major concerns. We have no further changes to make in response to this review, and we thank the reviewer again for the positive evaluation.

**Recommendations for the authors:**

**Reviewing Editor Comments:**
While the study presents a great methodological advancement, the phenomenological examples described could perhaps benefit from a little more mechanistic description/discussion. In particular, the functional effect of the V288A mutant is very novel. It could be useful to discuss whether this mutant impacts channel selectivity/conductance. It could be beneficial to also contrast the subunit dependence of V288A with that of the W434F mutant of the Shaker channel. In the latter, C-type inactivation gating is accelerated even when the mutant is present in a single subunit, which contrasts with the effect in V288A.

We greatly appreciate the editor’s and reviewers’ thorough and constructive evaluation, and we have revised the manuscript accordingly.

We added discussion with citation about the potential effect of V288A on selectivity (lines 348349). We also added the reported stoichiometric effects of mutations in Shaker and hERG1 channels on C-inactivation in discussion (lines 336-351). From these studies and our findings with V288A in BK channels, it is interesting to note that the stoichiometric effects of these mutations varies and those located near or within selectivity filter signature exhibited an all-or-none effect in both hERG1 and BK channels.

The authors might also want to consider performing and showing immunoblots with the alpha_deltaM fragment co-expressed with the other channel fragments. Together with the GFP tag, this alpha_deltaM would perhaps be a ~90 kDa protein. It should be captured by anti-V5 IP and resolved on an SDS-PAGE gel (at least with the quad construct).

We added supplemental data (Fig.1 – figure supplement 1) to show co-expression and co-IP of the α^ΔM^-GFP construct and a FLAG-tagged α_M_ construct. The α^ΔM^-GFP displayed right size on SDS-PAGE. It is of note that the single unit α_M_ construct tended to oligomerize even under denatured condition on SDS-PAGE.

For Figure 4, providing details about the inter-pulse intervals and interpulse holding voltage would be helpful. I was not able to find this information in the methods or text.

The inter-pulse intervals and holder voltage are now added in Fig. 4 legend (line 638).

**Reviewer #1 (Recommendations for the authors):**
(1) Submitted papers should have page numbers to facilitate reviewing.

Both page and line numbers are added.

(2) The designation of the various channel types, such as BKα and BKαM should be identical in the text and figures, so either drop BK in the text or add BK in the figures. Maybe drop BK in the text, as it is known that BK channels are the topic of this study.

We appreciate the suggestion to be consistent in text and figures. We have dropped “BK” for “BKα_M_” throughout the text.

(3) "Single Boltzmann fits of G-V curves" would be consistent with a homogenous channel population but do not necessarily suggest a single homogenous channel population of BK channels, as was shown by Geng et al. (2023) (https://doi.org/10.1085/jgp.202213302) where the G-V curve for simultaneous expression of five BK channel types with different V1/2s for each channel type was well approximated by a single Boltzmann function. The dogma that a single Boltzmann fit suggests one channel type needs to be reset. So wave a red flag here: whereas a single Boltzmann fit is consistent with a single channel type, it does not establish a single channel type nor even suggest a single channel type.

We fully agree that a good Single Boltzmann fit doesn’t mean homogenous channel population. We have changed “suggesting” to “consistent with” (line 203) and “reflecting” to “agreeing with” (line 205).

(4) Geng et al. (2023) demonstrated that the pore mutation G375R in BK channels gave a left shift in activation linearly related to the number of WT subunits replaced with mutant subunits. This should incremental shift in activation for G375R should be mentioned, as it is consistent with the incremental effects of the L312A deep pore mutation on activation as reported by the authors in their Figure 3D.

We appreciate the pointing-out of this highly relevant publication. We have now included this reference and discussed together with L312A mutation (lines 309-313).

(5) I went back and looked at the Lingle laboratory papers on the gamma subunit. An additional sentence or two on what the Lingle lab found and didn't find would be useful here for readers.

In the Introduction, we have listed the Lingle lab’s findings and the limitations of their experimental methods that warrants the development of a concatenated construct method as proposed in this study (lines 84-88). We prefer to not discuss further in the Discussion as it will be redundant.

(6) For the two examined mutations L312A and V288A, include in the Methods a 21 amino acid sequence for each mutation with the amino acid to be mutated (L or V) in the center, with beginning and end numbering at the beginning and end of each list. This will allow the reader/experimenter to readily locate the mutated residue on their BK amino acid sequences, which may have different numbering than U11058. Interestingly, for the so-called canonical sequence Q12791 · KCMA1_HUMAN that I found in UniProt starting with U11058, there is an L312, but I found no V288, but an F288. Am I doing this correctly? Do I have the correct sequence/isoform? The only sure way to identify an AA is with an extensive pre and post-sequence so that the chance of misidentification approaches zero.

We verified that the listed Gene Bank IDs of U11058 for cDNA and AAB65837 for protein should point to the right sequences. In the section of Results, we have now included the peptide sequences of the selectivity filter signature motif and part of the S6 TM where V288 and L312A are located, respectively (lines 179 and 220).

**Reviewer #2 (Recommendations for the authors):**
The different stoichiometry of the gamma subunit and the mutations in regulating channel function raise important questions. For instance, what are the structural and energetic bases for their different stoichiometric requirements? Does the structure motif, such as the selectivity filter or deep pore, act as a unit? Or does a specific residue, such as V288 or L312, act individually to determine the different stoichiometric requirements? What molecular interactions are involved for these residues and subunit to influence the cooperativity among the four alpha subunits in channel function? Some of these questions are discussed in the manuscript, but it may help the readers to clarify what aspects of the mechanistic bases for the findings in this manuscript are known and what aspects remain to be studied.

We agree that these are all important questions. We have now cited more previous studies on C-inactivation in other K^+^ channels and on deep pore mutations in BK channels in terms of subunit stoichiometry (lines 336-351). The results appear to be consistent, suggesting shared properties among residues within the selectivity filter motif or among residues in deep pore region.

Some minor comments are as follows.(1) Page 7, 2nd paragraph: "Page 2B" change to "Page 3B"? Also, "delay in deactivation" is not precise. The term "Delay" in channel kinetics has a specific meaning, and the use of this word here causes some confusion. The authors may want to delete "substantial delay in deactivation evident as a”.

Corrected by changing Fig. 2B to Fig. 3B and deleting “a substantial delay in deactivation evident as” (line 191).

(2) Page 9, 1st paragraph: "used in the voltage protocol used". Drop one of the instances of used".

Corrected by deleting the first “used” (line 246).

(3) Page 12, 1st paragraph: "Nonetheless, the tight inter-subunit cooperativity observed at the selectivity filter makes it a plausible candidate for serving as the activation gate, a property not yet demonstrated for the lower S6 segment." This seems to be an interesting idea. However, it is not clearly explained. The authors may want to clarify how the cooperativity is related to the activation gate.

We have now added a sentence with citations to discuss the requirement of intersubunit cooperativity for an activation gate to function (lines 354-357).

Other major changes: We updated immunoblot figures Fig1C and Fig2C for better presentation.